# Microbiota Transplantation in Individuals with Type 2 Diabetes and a High Degree of Insulin Resistance

**DOI:** 10.3390/nu16203491

**Published:** 2024-10-15

**Authors:** Ana María Gómez-Pérez, Araceli Muñoz-Garach, Agustín Lasserrot-Cuadrado, Isabel Moreno-Indias, Francisco J. Tinahones

**Affiliations:** 1Department of Endocrinology and Nutrition, The Biomedical Research Institute of Malaga, and Platform in Nanomedicine (IBIMA-BIONAND Platform), Virgen de la Victoria University Hospital, University of Malaga, 29016 Malaga, Spain; anamgp86@gmail.com (A.M.G.-P.); fjtinahones@uma.es (F.J.T.); 2CIBER in Physiopathology of Obesity and Nutrition (CIBEROBN), Carlos III Health Institute, 28029 Madrid, Spain; aracelimugar@gmail.com; 3Department of Endocrinology and Nutrition, Granada Biosanitary Research Institute of Granada (ibs.Granada), Virgen de las Nieves University Hospital, 18012 Granada, Spain; 4Andalusian Research, Development, and Innovation Plan. CTS 367, University of Granada, 18071 Granada, Spain; lasserrot@ugr.es; 5Faculty of Medicine, University of Málaga, 29071 Malaga, Spain

**Keywords:** type 2 diabetes, insulin resistance, probiotic, microbiota, fecal microbiota transplantation

## Abstract

Background/Objectives: The objective of this study was to determine the results of fecal microbiota transplantation (FMT) from healthy lean subjects in patients with type 2 diabetes (T2D); Methods: We designed a phase II, randomized, single-blind, parallel-arm clinical trial. Twenty-one subjects (12 men [57.1%] and 9 women [42.9%]), who had previously signed an informed consent were randomized to FMT from lean donors, a probiotic (*Lactobacillus delbrueckii* spp. *bulgaricus LB-14*), or placebo. Mean age at baseline was 62.5 ± 5.8 years and mean body mass index (BMI) at baseline was approximately 32.4 ± 2.4 kg/m^2^. Anthropometric measures, biochemical variables, oral glucose tolerance test (OGTT), and a stool microbiota analysis were performed (baseline, 4 and 12 weeks). The trial was conducted following the *Declaration of Helsinki*, *Good Clinical Practice Guides* (CPMP/ICH/135/95) and the current Spanish legislation regarding clinical trials (RD 223/2004).; Results: FMT changes occurred at the expense of the species found in the donor. No differences in weight, body mass index, HbA1c, or the results of the OGTT for glucose and insulin were found between groups after the intervention, although a decrease in uric acid was observed in the probiotic group (−0.5 mg/dL; *p* = 0.037) and a mild increase in HbA1c in the FMT group (+0.25%; *p* = 0.041); Conclusions: In our sample, neither FMT from healthy and lean donors nor a probiotic were effective in improving insulin sensitivity and HbA1c in patients with T2D.

## 1. Introduction

In recent decades, studies on the involvement of the gut microbiota in non-communicable diseases have been a constant, with increasing evidence of their involvement in obesity and diabetes. Therefore, it is not surprising that microbiota therapies such as probiotics or fecal microbiota transplantation (FMT) have been raised as a possible therapeutic tool in diverse diseases, including obesity, diabetes, and other metabolic diseases, as well as inflammatory bowel disease and infectious diseases [1,2,3,4,5]. However, FMT is currently only approved in the USA for the treatment of recurrent and refractory Clostridium Difficile infection [6]. Regarding its use in the treatment of type 2 diabetes (T2D), few studies are available, showing mild improvements in glycemic control. For example, Ding et al. found an improvement in glycated hemoglobin (HbA1c), blood glucose, and uric acid 12 weeks after FMT from healthy donors. The FMT responders showed higher levels of the family *Rikenellaceae* and the genus *Anaerotruncus* at baseline compared to non-responders [7]. However, most studies did not identify any positive outcomes in patients with obesity or insulin resistance, as evidenced by a recent meta-analysis encompassing six studies with 154 participants. The authors observed a minimal impact on HbA1c in the short term, which subsequently diminished over time, without any discernible enhancement in other anthropometric or biochemical parameters [8]. Therefore, there is a need for more data from randomized controlled trials to clarify the role of FMT, and probiotics, in metabolic diseases.

The main objective of this study was to determine whether changes in the microbiota after FMT from healthy lean subjects or after treatment with a probiotic applied to patients with T2D and high insulin resistance produce changes in insulin sensitivity as assessed by the Homeostatic Model Assessment insulin resistance index (HOMA-IR) and the HbA1c.

## 2. Materials and Methods

This Phase II, randomized, single-blind, parallel-arm clinical trial was performed at Virgen de la Victoria University Hospital in Málaga (Spain), as a sub-study of the project “*Epigenetic modifications and microbiota in the genesis of adipose tissue dysfunction and insulin resistance*” (EPIGEN-MICROBIOTA), and was approved by our local Ethics Committee. The trial was conducted following the *Declaration of Helsinki*, *Good Clinical Practice Guides* (CPMP/ICH/135/95) and the current Spanish legislation regarding clinical trials (RD 223/2004). This trial is registered under the name “Epigenetic and Microbiota Modifications” with the code NCT05076656. Subjects included in the trial met the following criteria: T2D treated with metformin; body mass index (BMI) 30–40 kg/m^2^; age 30–70 years old; HOMA-IR > 2; signed informed consent. Exclusion criteria included the following: treatments for diabetes different from metformin; previous history of cholecystectomy or treatment with antibiotics or probiotics in the 3 months prior to inclusion. Twenty-one subjects were randomized to three different arms: FMT orally via lyophilization in a capsule, following the protocol proposed by Kao et al. [2]; therapy with a probiotic (PB) included in the list of *Generally Recognized as Safe bacteria* (GRAS) by the Food and Drugs Administration (FDA) (https://www.fda.gov/Food/IngredientsPackagingLabeling/GRAS/; accessed on 15 April 2019) comprising *Lactobacillus delbrueckii* spp. *bulgaricus* LB-14 with 25 × 10^9^ CFUs/pill once daily; and placebo administration (a product similar to the probiotic and FMT, but without their contents). The FMT came from two healthy and lean donors, aged between 40 and 65 years, BMI < 25, without metabolic syndrome criteria. The donor’s fecal samples were processed following the recommendations by Kao et al. [2]. The investigational therapy was administered in a single dose on the second day of visit 1, following a bowel lavage the previous night. Participants received the capsules in the office and took them immediately. They were then observed for approximately two hours to assess for any immediate adverse effects. All participants were asked to maintain their usual diet and physical activity throughout the study.

Study subjects were assessed at baseline, and at 4 weeks and 12 weeks, with anthropometric measures, a blood test, and a 75 g oral glucose tolerance test (OGTT) for glucose and insulin. Stool samples were collected at each visit for microbiota analysis through the sequencing of the ribosomal 16S rRNA gene following the methodology of Gómez-Pérez et al. [9] and Díaz-Perdigones et al. [10].

To assess the impact of the intervention on anthropometric variables, HOMA-IR, and HbA1c, a statistical analysis was performed through a study of the frequency distribution of the qualitative variables, and the median and interquartile range of the quantitative ones. To compare the hypotheses, we used non-parametric tests for continuous variables: for comparison between two groups, the Mann–Whitney test; for a comparison between before and after in the same group, we used the Wilcoxon test; and for comparisons between several groups, we used the Kruskal–Wallis test. Chi2 tests were used to compare the hypotheses regarding qualitative variables. The rejection level of Ho was 0.05 for one or two tails, according to the hypothesis presented at the beginning of the study.

With respect to changes in the microbiota, profiling was performed using the tool Ion Reporter (Ion Reporter Software 5.12, Thermofisher; Life Technologies Holdings Pte Ltd., Singapore 739256), as well as clustering with the reference base Greengenes version 13_5 at 99% identity and the curated MicroSEQ^®^ 16S Reference Library V2013.1 at the species level (further information in Appendix A). Feature tables were used to search for species shared between the donor and receptor using Venn diagrams.

## 3. Results

A total of 21 subjects were included: 57.1% (*n* = 12) were men and 42.9% (*n* = 9) were women. Seven individuals were assigned to each group, although two participants (one from the FMT and one from the placebo group) withdrew from the study after the intervention, for reasons unrelated to the investigational therapy. At baseline, the three groups were homogeneous, without statistically significant differences between characteristics (Table 1). The median weight of the placebo group was 90 (80.5–100.9) kg, while median weight was 92 (79–98.5) kg for the PB group and 83.8 (77.3–89) kg for the FMT group (*p* = 0.675). Baseline glucose levels were 99.5 (87.8–111.5) mg/dL in the placebo group, 128 (99–134) mg/dL in the PB group, and 110 (99–129.5) mg/dL in the FMT group (*p* = 0.314). Regarding HbA1c, the figures were 6.5% (6.2–7.7%) in the placebo group, 7% (5.9–5.3%) in the PB group, and 6.6% (6.1–6.9%) in the FMT group (*p* = 0.186). A comparable trend was noted for HOMA-IR, with a median of 2.3 (1.7–4.5) in the placebo group, 3 (0.6–5.3) in PB, and 3 (1.2–3.8) in FMT (*p* = 0.091). No differences were found in the HOMA-beta or Matsuda indices at baseline.

After the *12*-week intervention, there were no significant differences between groups regarding anthropometric measures: median weight in the placebo group was 90 kg (IQR 80.5–100.9), in PB group, median weight was 92 kg (IQR 79–98.5), and in the FMT group, median weight was 83.8 kg (IQR 77.3–89) (*p* = 0.529). No difference was found in metabolic status (median HbA1c 6.2% (IQR 5.9–6.4%) in the placebo group; 6.2% (IQR 6–6.9%) in the PB and 6.6% (IQR 6.1–6.9%) in the FMT group [*p* = 0.225]), in HOMA-IR, HOMA-beta, and Matsuda indices (Table 1), or in the results of the OGTT regarding glucose and insulin (Figure 1) after the intervention.

Twelve weeks after the intervention, no differences were found in any clinical or biochemical variable in the placebo group. In the PB group, the results were similar to those of the placebo group, but with a significant decrease in uric acid 12 weeks after the intervention, from 7 mg/dL (IQR 5.9–7.4 mg/dL) to 6.5 mg/dL (IQR 5.3–6.8 mg/dL) (*p* = 0.027). Finally, in the FMT group, there was a mild worsening of the HbA1c 3 months after the intervention, with a median at baseline of 6.6% (IQR 6.1–6.9%), and after FMT of 6.9% (IQR 6.3–7%) (*p* = 0.041). No other significant difference was found in this group (Table 1).

The diversity and profile of the gut microbiota did not significantly differ between treatments or sampled points. However, within experimental treatments, the PB lost the fewest species compared to baseline (Figure 2a). Interestingly, changes in the FMT group were at the expense of an increase in the species found in the donor (Figure 2b). Finally, *Lactobacillus fermentum* was only found in one sample of PB.

During the study period, no mild or serious adverse events related to the investigational therapies were reported.

## 4. Discussion

We found that FMT from healthy and lean donors was not effective in our sample to improve insulin sensitivity and HbA1c in patients with T2D, in line with previously reported results. Yu et al. [11] performed a 12-week, double-blind clinical trial in 24 adults with obesity and a high risk of T2D. The participants were randomized to receive either weekly oral FMT capsules from healthy and lean donors or placebo capsules for 6 weeks. After 12 weeks, they found no significant differences between groups in terms of either anthropometric measures or metabolic parameters such as HOMA-IR or HbA1c. Therefore, as was found in our study, although the procedure was safe and changes in the microbiota’s composition were observed, they were not enough to induce a positive effect in terms of metabolic function. One metanalysis of six studies with 154 participants assessed the role of FMT in the treatment of obesity and metabolic syndrome, finding a minimum decrease in HbA1c 6 weeks after the intervention that disappeared in the long-term, as well as in our study, without an improvement in anthropometric parameters, glucose levels, or insulin sensitivity compared to placebo [8]. Another study in individuals with T2D comparing a dietary intervention with a group combining this intervention with FMT (eight participants per group) reported a significant decrease in blood glucose and weight loss after 90 days in the diet group but not in the FMT group [12].

Regarding weight, some studies were conducted to assess the impact of autologous FMT or FMT from lean donors on weight loss and the prevention of weight being regained. The results are contradictory, with some of them reporting a possible influence of the FMT on weight loss or weight maintenance [13] and others in line with our results, showing no changes in BMI after the intervention [4] or no differences in the percentage of weight loss in the short- and long-term after bariatric surgery [14]. However, in the study reported by Rinott et al., participants underwent a dietary intervention for weight loss before the autologous FMT; therefore, changes in microbiome composition and weight maintenance could have been more influenced by the dietary modifications [13].

Some other studies are in contrast with our results, but the clinical impact of the intervention is doubtful. Ding et al. performed a transendoscopic enteric tube FMT in 17 individuals with T2D without a control group and though they found a mild decrease in HbA1c, glucose levels, and uric acid 12 weeks after the intervention (*p* < 0.01), they also reported individual variability, so the changes cannot be directly related to the intervention [7]. Siew et al. reported the effects of repeated FMT every 4 weeks for 24 weeks combined with dietary modifications or alone, on the microbiota modifications of subjects with T2D and obesity, but they found no effects on glycemic control or weight [15]. A recent metanalysis including four randomized control trials found that, compared with the results obtained for non-FMT groups, combined FMT treatment may reduce plasma glucose levels in patients with T2D, and compared with single FMT treatment, combined FMT could also improve triglycerides and total cholesterol levels. Even single FMT treatment may decrease HOMA-IR, triglycerides, and HDL levels [16].

The procedure was safe in the majority of the studies but some reported adverse events following FMT in patients with obesity, mainly gastrointestinal symptoms [15,17].

Finally, in our study, gut microbiota showed no statistically significant changes, although donor species increased at the expense of the receptor ones, indicating a possible restoration [18]. However, in the probiotic group, *Lactobacillus fermentum* was not widely observed in the volunteer samples, although this does not imply that it could be observed in the intestine mucosa [19]. In a randomized, placebo-controlled, double-blind clinical trial, Wilson et al. conducted an FMT in 87 adolescents with obesity. The FMT consisted of capsules containing the fecal microbiota of four lean same-sex donors. They observed that although multi-donor FMT resulted in a sustainable alteration in the structure and function of the gut microbiome, the dominant strain engraftment was attributed to two specific donor microbiomes (one female and one male). These microbiomes were distinguished by their high microbial diversity and a high *Prevotella/Bacteroides* ratio. Similarly, although they attempted to standardize the dose and origin of FMT, there was considerable variation in the engraftment of donor strains among recipients. This suggests there is also a possible impact of the host environment on the engraftment of FMT, which could partially explain the different results [20].

## 5. Conclusions

Our findings indicate that the intervention involving FMT and probiotic supplementation did not result in improved glucose tolerance in individuals with type 2 diabetes. Additionally, these interventions did not lead to notable alterations in the gut microbiota among the participants. However, there was an observed increase in donor species, which came at the expense of the recipient species.

The main limitations of our study are the small sample size and the very specific study population, which was limited to patients treated with metformin, who generally had a good initial metabolic controfsuppleml. On the other hand, we were not able to use the gold standard for the assessment of insulin resistance because the euglycemic–hyperinsulinemic clamp is a difficult test and we did not have the necessary media to perform it. This makes it difficult to generalize our conclusions.

## Figures and Tables

**Figure 1 nutrients-16-03491-f001:**
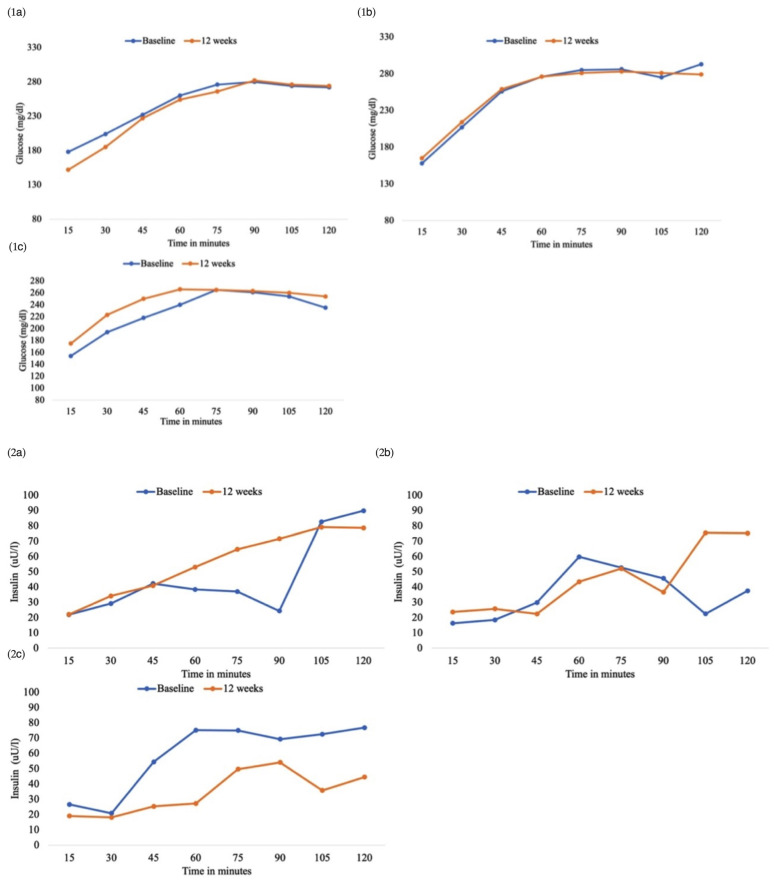
(**1a**) Results of the 75 g oral glucose tolerance test (OTTG) regarding glucose in the placebo group. (**1b**) Results of the 75 g oral glucose tolerance test (OTTG) regarding glucose in the probiotic treatment group. (**1c**) Results of the 75 g oral glucose tolerance test (OTTG) regarding glucose in the fecal microbiota transplantation (FMT) group. (**2a**) Results of the 75 g oral glucose tolerance test (OTTG) regarding insulin in the placebo group. (**2b**) Results of the 75 g oral glucose tolerance test (OTTG) regarding insulin in the probiotic treatment group. (**2c**) Results of the 75 g oral glucose tolerance test (OTTG) regarding insulin in the fecal microbiota transplantation (FMT) group.

**Figure 2 nutrients-16-03491-f002:**
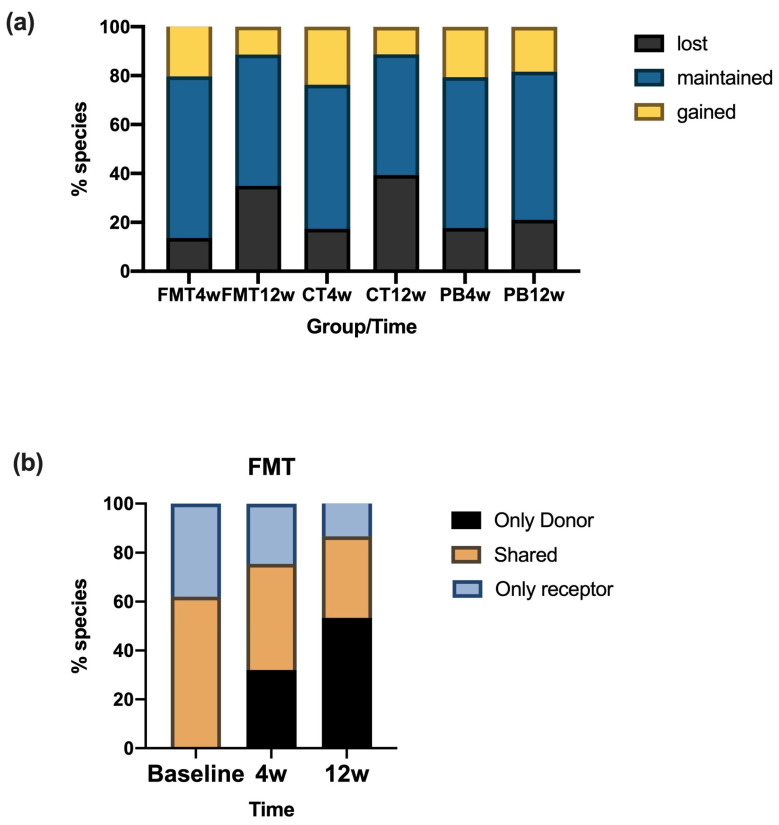
(**a**) Number of bacteria species that were lost, maintained, or gained with respect to the baseline profile with each of the interventions: fecal microbiota transplant (FMT), probiotic intervention (PB), or placebo groups at 4 weeks or 12 weeks. (**b**) Within the FMT intervention, the number of bacteria species shared between the donor and the receptor, the number of species only found in the donor, and the number of species only found in the receptor of the intervention in relation to their respective donors at different time points.

**Table 1 nutrients-16-03491-t001:** Comparative analysis of main variables for each intervention group between baseline and 12 weeks after intervention.

	Placebo (*n* = 7)	Probiotic (*n* = 7)	FMT (*n* = 7)	*
Variables				
Weight (Kg), median (IQR)				
Baseline	90 (80.5–100.9)	92 (79–98.5)	83.8 (77.3–89)	*p* = *0.675*
12-weeks	85 (75–98.3)	93.5 (80–96)	84.5 (74.9–89.5)	*p* = *0.457*
**	*p* = *0.345*	*p* = *0.397*	*p* = *0.917*	
Waist circumference (cm), median (IQR)				
Baseline	108 (102–112.3)	107 (105–113)	105 (98.5–112.3)	*p* = *0.858*
12-weeks	113 (101.3–142)	110 (105–114)	103.5 (95.3–113.5)	*p* = *0.439*
**	*p* = *0.138*	*p* = *0.750*	*p* = *0.400*	
BMI (Kg/m^2^), median (IQR)				
Baseline	33.7 (31.8–36.3)	31.3 (30.4–31.9)	31.1 (30.3–32.9)	*p* = *0.414*
12-weeks	33.2 (30.3–35.2)	30.7 (30.5–32.4)	31.6 (29–33.8)	*p* = *0.625*
**	*p* = *0.345*	*p* = *0.398*	*p* = *0.917*	
SBP (mmHg), median (IQR)				
Baseline	134 (124–141)	129 (117–153)	127 (120–134.3)	*p* = *0.251*
12-weeks	138 (123.5–158)	131 (128–149)	118.5 (109.3–135.8)	*p* = *0.220*
**	*p* = *0.078*	*p* = *0.352*	*p* = *0.249*	
DBP (mmHg), median (IQR)				
Baseline	87.5 (78.8–91)	80 (74–87)	84 (76–90)	*p* = *0.363*
12-weeks	83 (80.5–92.5)	83 (77–93)	82.5 (70.3–88.3)	*p* = *0.668*
**	*p* = *0.684*	*p* = *0.115*	*p* = *0.916*	
Fasting glucose (mg/dL), median (IQR)				
Baseline	99.5 (87.8–111.5)	128 (99–134)	110 (99–129.5)	*p* = *0.314*
12-weeks	113 (100.5–154.5)	114 (103–115)	122.5 (99–143.5)	*p* = *0.719*
**	*p* = *0.138*	*p* = *0.128*	*p* = *0.115*	
HbA1c (%), median (IQR)				
Baseline	6.5 (6.2–6.6)	6.1 (5.9–6.7)	6.6 (6.1–6.9)	*p* = *0.557*
12-weeks	6.2 (5.9–6.4)	6.2 (6–6.9)	6.9 (6.3–7)	*p* = *0.225*
**	*p* = *0.273*	*p* = *0.236*	*p* = *0.041*	
Uric acid (mg/dL), median (IQR)				
Baseline	5.5 (3.8–7.7)	7 (5.9–7.4)	5.4 (4.8–6)	*p* = *0.186*
12-weeks	5.4 (4.6–6.3)	6.5 (5.3–6.8)	5.5 (3.7–6.1)	*p* = *0.206*
**	*p* = *0.686*	*p* = *0.042*	*p* = *0.750*	
HOMA IR, median (IQR)				
Baseline	2.3 (1.7–4.5)	3 (0.7–5.3)	3 (1.2–3.8)	*p* = *0.091*
12-weeks	4.5 (4–11.4)	2.5 (0.7–3.8)	3.4 (0.9–4.2)	*p* = *0.112*
**	*p* = *0.118*	*p* = *0.866*	*p* = *0.600*	
HOMA-beta, median (IQR)				
Baseline	81.2 (77.1–118.7)	63.4 (22.8–87.9)	66.5 (35.9–100.3)	*p* = *0.109*
12-weeks	134.5 (100.2–175.2)	62.3 (31.4–100.3)	51.2 (28.3–69.9)	*p* = *0.085*
**	*p* = *0.813*	*p* = *0.468*	*p* = *0.313*	
Matsuda Index, median (IQR)				
Baseline	3.6 (1.7–8.6)	5.5 (2–8.8)	6.4 (4.7–13.5)	*p* = *0.564*
12-weeks	1.9 (1.4–2.3)	2.3 (1.8–3.2)	2.1 (1.7–2.7)	*p* = *0.657*
**	*p* = *0.250*	*p* = *0.875*	*p* = *0.500*	

IQR: interquartile range; BMI: body mass index; SBP: systolic blood pressure; DBP: diastolic blood pressure; HOMA IR: Homeostatic Model Assessment for Insulin Resistance; HOMA-beta: Homeostatic Model Assessment beta or Beta Cell Activity Index; * *p*-value at baseline and 12 weeks between intervention groups; ** *p*-value at 12 weeks intragroup.

## Data Availability

The data set generated, as well as the complete protocol and methodology, are available upon reasonable request from the corresponding author.

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
