# Peer review of "Microbiota Transplantation in Individuals with Type 2 Diabetes and a High Degree of Insulin Resistance"

_nutrients, 2024, doi:10.3390/nu16203491_

Round 1

Reviewer 1 Report

Comments and Suggestions for Authors

The authors provide results from an RCT investigating the impact of FMT or probiotic supplementation on body weight and metabolism in patients with obesity.

The overall rationale is clear.

Abstract: More info about the cohort structure (age, sex, BMI, ...) is warranted.

Introduction: Rather short; in particular lines 47-50 should be vastly extended and substantiated by sufficient references.

Methods:

Study title and abbreviation differ from the entry at clinicaltrials.gov.

Inclusion criteria do not perfectly match the ones submitted to clinicaltrials.gov; please clarify.

Statistics lack definition of the primary outcome and a power calculation.

Statistical analysis requires Kruskal-Wallis-Tests on the changes from baseline to week 4 or week 12, not on the measurements of weeks 4 and 12 itself. Otherwise comparison of week 4 and 12 is biased by minor baseline differences.

oGTT data should be analysed by ANOVAs.

Statistics leave it open, if you did ITT, as-treated or PP analysis.

Did you discontinue metformin treatment prior to the main visits? Otherwise, oGTTs will provide biased data.

The dosage of probiotics is missing, how many CFUs did it contain?

The three treatments were administered just once or once daily over the entire time course?

Did the patients receive any additional recommendations, restrictions or guidelines for changes or maintenance of original diet and/or physical activity?

Why did you not include 30 patients as originally planned?

Why did you abstain from blinding personnel and statisticians?

Results:

Line 92: Please provide the actual data for that comparison. (ref. to table 1, but it needs the p-values for baseline comparison, too)

Line 93-100: It is unclear, if this is 4-weeks or 12-weeks data.

If you did an oGTT, postprandial IR (Matsuda, Stumvoll ...) and beta cell function (HOMA-beta, disposition index) are of interest.

Table 1: "Glucose" is fasting plasma glucose, isn't it? What about 2h glucose?

Fig 3B: FMT: How can the gut microbiome at baseline contain bacteria that are only found in the donor? Sampling took place before FMT, did it?

Discussion: can only be evaluated after major revision

However, I already notice, that you only referenced 14 papers in total. In the light of the extensive amount of literature on the topic, I encourage you to provide a better comparison of your data with the existing plethora of data.

Comments on the Quality of English Language

minor changes needed

Author Response

We would like to express our sincere gratitude for all your constructive comments and recommendations. We really appreciate your help and advice, which have allowed us to improve our manuscript.

We have provided the replies to Reviewer 1 and 2´s comments in the following section and have highlighted changes in the manuscript in yellow.

We hope that our revised manuscript may now be considered acceptable for publication in the journal. Nevertheless, we are of course willing to revise it further according to any other suggestions or concerns raised by the Editor or the Reviewers.

Yours sincerely,

The authors.

Reviewer Comments to Authors

Reviewer 1

The authors provide results from an RCT investigating the impact of FMT
or probiotic supplementation on body weight and metabolism in patients
with obesity.

Comment 1: The overall rationale is clear.

Abstract: More info about the cohort structure (age, sex, BMI, ...) is
warranted.
Response 1: We appreciate the reviewer's feedback and apologize for the lack of information in the abstract. We have included some of this information in the abstract (lines 25-28).

Comment 2: Introduction: Rather short; in particular lines 47-50 should be vastly
extended and substantiated by sufficient references.

Response 2: thank you very much for your kind comment and suggestion. Indeed, the introduction is brief, accordingly with the kind of manuscript, a brief report, to address the results of a pilot study. However, we have augmented the data presented in this section to enhance the overall comprehensiveness of the evidence in the field (lines 43-57), though maintaining it short according to the article type, a brief report.  

Comment 3: Methods: Study title and abbreviation differ from the entry at
clinicaltrials.gov.

Inclusion criteria do not perfectly match the ones submitted to
clinicaltrials.gov; please clarify.

Response 3: Thank you for your kind comments. We have verified that the title and abbreviation of the study do not correspond to those registered in ClinicalTrials.gov. This discrepancy is because this trial is part of the main project, as it was approved by the Ethics Committee, but the trial was the only part registered in ClinicalTrials.gov under the name “Epigenetic and Microbiota Modifications”. This information has been included within the text for clarification.

With regards to the inclusion criteria, the first version of the protocol was amended to reflect a change of cut-off point, considering recruitment difficulties. As the normal HOMA-IR value ranges from 0.7-1.9 (PMID: 24530467, PMID: 20721461, PMID: 18796548), we included patients with elevated insulin resistance >2.

Comment 4: Methods: Statistics lack definition of the primary outcome and a power
calculation.
Statistical analysis requires Kruskal-Wallis-Tests on the changes from
baseline to week 4 or week 12, not on the measurements of weeks 4 and 12
itself. Otherwise comparison of week 4 and 12 is biased by minor
baseline differences. oGTT data should be analyzed by ANOVAs.
Statistics leave it open, if you did ITT, as-treated or PP analysis.

Response 4: The authors apologize for the insufficient information regarding the statistical analysis, stated in a general manner, mainly because of the brief report nature of the manuscript. As the referee suggest, the results from oGTT data were measured by Kruskal-Wallis test, instead of ANOVA, due to the non-parametric characteristics of the data. No differences were found regarding these parameters.

Comment 5: Did you discontinue metformin treatment prior to the main visits?
Otherwise, oGTTs will provide biased data.

Response 5: We appreciate your kind question. Metformin was not discontinued during the study. As we know, metformin can cause changes in insulin sensitivity and in the gut microbiota. Nevertheless, the aim was to evaluate changes in the various parameters analyzed, including OTTG. In addition, the treatment was maintained in the three groups, so we believe this does not confound the comparison between the different interventions. Conversely, it was deemed unethical to suspend effective treatment in the interest of avoiding a potential worsening of metabolic control. Nevertheless, the use of metformin is considered a limitation of our study (line 185).

Comment 6: The dosage of probiotics is missing, how many CFUs did it contain?

Response 6: The authors apologizes for the absence of this important information. The dosage was 25 x 109 CFUs/pill once daily.  This information has been introduced within the main text (lines 80-81).

Comment 7: The three treatments were administered just once or once daily over the
entire time course?

Response 7: We thank the reviewer for their feedback and apologize for the lack of clarity regarding the FMT protocol in the manuscript. The investigational therapy was administered in a single dose on the second day of visit 1, following a bowel lavage the previous night. Participants received the capsules in the office and took them immediately. They were then observed for approximately two hours to assess for any immediate adverse effects. On the other hand, probiotic was administered once a day during 1 month. This information has been introduced within the main manuscript.

Comment 8: Did the patients receive any additional recommendations, restrictions or
guidelines for changes or maintenance of original diet and/or physical
activity?
Response 8: Thank you for your pertinent question. As diet and exercise are known to induce important changes in gut microbiota, and the main objective was to analyze the effect of FMT, in order to not introduce more bias, all participants were asked for not changing their usual lifestyle throughout the period of study.  This information has been introduced within the main text.

Comment 9: Why did you not include 30 patients as originally planned?

Response 9: thank you for your question. The initial sample size was 10 participants per group. However, with the COVID19 pandemic and the health emergency in our country, recruitment and follow-up of the patients were hampered, because of the different measurements adopted by our government as well as due to patients' fear of going to a hospital. In that moment we performed a first analysis of the results, and this analysis showed no efficacy for the therapies studied, and therefore the originally calculated sample was not completed. However, we consider that the publication of these negative results is very necessary for the FMT science, to continue advancing for translational research.

Comment 10: Why did you abstain from blinding personnel and statisticians?

Response 10: thank you for your question. Although it is a good point to consider for future interventions, as FMT pills manipulation carried extra care because of its nature, such as the storage in -80ºC, we preferred to have total control over the process. Thus, with the experience gained with this trial, this point will be considered for future studies.

Comment 11: Line 92: Please provide the actual data for that comparison. (ref. to
table 1, but it needs the p-values for baseline comparison, too)

Response 11: Thank you for your valuable input. Table 1 has been updated to include the p-value for baseline differences between intervention groups, and the most relevant data has been incorporated into the main text (lines 114-121).

Comment 12: Line 93-100: It is unclear, if this is 4-weeks or 12-weeks data.

Response 12: Thank you for bringing this to our attention. We apologize for the confusion caused by the lack of clarity in the information provided. In the text, we focused on the results at the end of the intervention at 12 weeks. We have now amended this in the text (line 123) to specify this time frame.

Comment 13: If you did an oGTT, postprandial IR (Matsuda, Stumvoll ...) and beta
cell function (HOMA-beta, disposition index) are of interest.

Response 13:  Thank you for your comments and suggestions. Following your recommendations, we calculated the Matsuda index to assess postprandial insulin resistance and HOMA-beta to assess beta cell function. We found no significant differences between the intervention groups, either at baseline or 12 weeks after the intervention. Similarly, no significant within-group differences were found when comparing results at 12 weeks with baseline. In order not to complicate the manuscript and not to lose the character of a brief report, we have not included these results in the manuscript. However, a detailed summary of the results of these analyses is provided below:

Matsuda Index: At baseline, there were no statistically significant differences between groups in the Matsuda Index with a median of 3.56 (IQR, 1.71-8.64) in the placebo group, 5.51 (IQR, 2-8.82) in the PB group and 6.41 (IQR, 4.68-13.53) in the FMT group (p0.564). Similar results were seen at 12 weeks with a median of 1.94 (IQR, 1.38-2.28) in the placebo group, 2.25 (IQR, 1.77-3.20) in the PB group and 2.09 (IQR, 1.72-2.71) in the FMT group (p0.657).

A comparison of within-group differences from baseline to 12 weeks yielded the following p-values: p0.250 for the placebo group, p0.875 for PB, and p0.500 for FMT.

HOMA-beta: There were no statistically significant differences between groups at baseline or at the end of the intervention (12 weeks). At baseline, the median was 81.21 (IQR, 77.10-118.70) in the placebo group, 63.37 (IQR, 22.81-87.89) in the PB group, and 66.46 (IQR, 35.89-100.25) in the FMT group (p0.109). At 12 weeks, the results were as follows: 134.53 (IQR, 100.18-175.18) for placebo, 62.30 (IQR, 31.44-100.33) for PB and 51.23 (IQR, 28.3-69.88) for FMT (p0.085). 

The results of the within-group Wilcoxon analysis revealed no statistically significant differences for this index, with p-values of 0.813 for the placebo group, 0.468 for the PB group, and 0.313 for the FMT group.

However, we are opened to its inclusion in the manuscript if the reviewer deems it appropriate.

Comment 14: Table 1: "Glucose" is fasting plasma glucose, isn't it? What about 2h
glucose?

Response 14: Thank you for your question. In response, it can be confirmed that the glucose in question is the fasting plasma glucose, as has been indicated in the table 1 to clarify this point. Additionally, the glucose at 2 hours after the OGTT is shown in Figure 1, comprising panels (a), (b) and (c).

Comment 15: Fig 3B: FMT: How can the gut microbiome at baseline contain bacteria
that are only found in the donor? Sampling took place before FMT, did
it?
Response 15: As evidence suggests, the gut microbiota profile is unique of an individual, being developed through its personal experiences. Although many bacterial are shared between different persons, others are characteristic of an individual. In this manner, the essence of an FMT would be to introduce donor characteristics into the receptor. The way in which this figure has been created was through the use of Venn diagrams to be able to observe the features that both, donor and receptor, were sharing at each moment. So, the fact that there were features only found in the donor does not mean that they are in the receptor, means that these features were not shared before FMT with the donor. A further explanation has been introduced into the text in order to clarify.  

Comment 16: Discussion: can only be evaluated after major revision.
However, I already notice, that you only referenced 14 papers in total.
In the light of the extensive amount of literature on the topic, I
encourage you to provide a better comparison of your data with the
existing plethora of data.

Response 16: We are truly grateful for your thoughtful comments and recommendations, which have helped us identify areas for improvement in the manuscript. We recognize that the discussion is relatively brief and that the references could be more extense. Given the nature of this brief report on the results of a small pilot study, we aimed to align the manuscript with the typical format for such reports. However, we have revised the discussion and included additional references to strengthen the message.

Comment 17: Comments on the Quality of English Language
minor changes needed.

Response 17: We would like to thank the reviewer for their recommendation and for their kind evaluation of our work. We have made the necessary corrections to the manuscript, including spelling and grammatical errors.

Final comments

Once again, we are grateful for the opportunity to revise and improve our manuscript.

We hope that our revised manuscript may now be considered acceptable for publication in the journal. Nevertheless, we are of course willing to revise it further according to any other suggestions or concerns raised by the Editor or the Reviewers.

Yours faithfully,

The authors.

Reviewer 2 Report

Comments and Suggestions for Authors

1. The frequency of the FMT is not clear: is it daily or weekly or bi-weekly treatment

2. The inclusion and exclusion criteria are valid. Need more clarification on the reason for selecting only T2DM patients receiving "Metformin" treatment (and excluding others) in discussion

3. Adverse effects of FMT - if any, found in your study

4. Figures need to be of higher resolution with increased font size - quite very challenging to see/read

Comments on the Quality of English Language

Must check and improve spellings: for example, replace "lactobacillus fermentuN" with "lactobacillus ferementuM" 

Also, replace "signifantative" with "significant"

There are grammatical mistakes as well 

Author Response

We would like to express our sincere gratitude for all your constructive comments and recommendations. We really appreciate your help and advice, which have allowed us to improve our manuscript.

We have provided the replies to Reviewer 1 and 2´s comments in the following section and have highlighted changes in the manuscript in yellow.

We hope that our revised manuscript may now be considered acceptable for publication in the journal. Nevertheless, we are of course willing to revise it further according to any other suggestions or concerns raised by the Editor or the Reviewers.

Yours sincerely,

The authors.

Reviewer Comments to Authors

Reviewer 2:

Comment 1: The frequency of the FMT is not clear: is it daily or weekly or
bi-weekly treatment.

Response 1: We thank the reviewer for their feedback and apologize for the lack of clarity regarding the FMT protocol in the manuscript. The investigational therapy was administered in a single dose on the second day of visit 1, following a bowel lavage the previous night. Participants received the capsules in the office and took them immediately. They were then observed for approximately two hours to assess for any immediate adverse effects.

The aforementioned information has been incorporated into the manuscript (lines 85-89).

Comment 2: The inclusion and exclusion criteria are valid. Need more
clarification on the reason for selecting only T2DM patients receiving
"Metformin" treatment (and excluding others) in discussion.

Response 2: thank you very much for your commentary. All patients included in the trial were being treated with metformin, as per the initial protocol, which was to include patients with good metabolic control. In our setting, these patients are typically treated with this medication as a standalone therapy, in accordance with the primary clinical practice guidelines, which recommend this drug as the preferred initial treatment option. The exclusion of other drugs from the inclusion criteria addresses this issue and ensures consistency, as most diabetes treatments can impact the microbiota. Given the pilot nature of the study with a limited sample size, it was preferable to have all participants with the same diabetes medication.

Comment 3: Adverse effects of FMT - if any, found in your study.

Response 3: thank you very much for your commentaries and questions. The occurrence of adverse events was closely monitored throughout the study in all three intervention groups, including the FMT group. The investigational therapies were not associated with any adverse effects, mild or severe, as detailed in lines 186-187 of the manuscript:

“During the study period, no mild or serious adverse events related to investigational therapies were reported”.

Comment 4: Figures need to be of higher resolution with increased font size -
quite very challenging to see/read.

Response 4: thanks for your kind commentary. The font and size of the figures have been updated to enhance readability and comprehension. Moreover, the figure has been separated into OGGT and microbiota data in two figures.

Comment 4: Comments on the Quality of English Language.

Must check and improve spellings: for example, replace "lactobacillus
fermentuN" with "lactobacillus ferementuM".  Also, replace "signifantative" with "significant".

There are grammatical mistakes as well.

Response 4: We would like to thank the reviewer for their recommendation and for their kind evaluation of our work. The manuscript has been reviewed for correct English spelling, and all suggested changes have been made. Additionally, all grammatical mistakes have been checked and corrected.

Final comments

Once again, we are grateful for the opportunity to revise and improve our manuscript.

We hope that our revised manuscript may now be considered acceptable for publication in the journal. Nevertheless, we are of course willing to revise it further according to any other suggestions or concerns raised by the Editor or the Reviewers.

Yours faithfully,

The authors.

Round 2

Reviewer 1 Report

Comments and Suggestions for Authors

The authors have revised their manuscript in accordance to the reviewer's suggestions.

Few minor points remain:

Abstract: Please shorten decimals in accordance to plausible raw data precision.

oGTT indices: Please include data for Matsuda, HOMA-Beta (=pre-prandial...) and disposition index (=...post-prandial insulin secretion capacity) in the manuscript.

Fig. 2B: This figure is still not plausible (to me). Already at baseline, patients in the FMT group (=receivers) seemed to share some gut microbiota with donors (which is plausible; orange bar), but you also claim, that they had other microbiota which were only present in donors (black bar), even though they did not yet receive FMT. So, if the black bar is not about shared species, how can receivers contain species, that are specific to donors and could not have yet reached the receivers? At baseline, there can only be orange and blue bars.

Comments on the Quality of English Language

minor changes needed

Author Response

We would like to thank you again for reviewing our work and for your comments and recommendations, which allowed us to improve our manuscript.

We have answered Reviewer 1's comments below and have submitted a new version of the manuscript with the suggested changes.

We hope that our revised manuscript can now be considered acceptable for publication in the journal. However, we are of course willing to revise it further according to any other suggestions or concerns raised by the editor or reviewers.

Sincerely yours, 

The Authors.

Reviewer Comments to Authors: 

Comment 1: Few minor points remain:

Abstract: Please shorten decimals in accordance to plausible raw data precision.

Answer 1: The decimals have been accordingly shorten. We also homogenized the decimals throughout the rest of the document.

Comment 2: oGTT indices: Please include data for Matsuda, HOMA-Beta (=pre-prandial...) and disposition index (=...post-prandial insulin secretion capacity) in the manuscript.

Answer 2: Thank you for your recommendation. The data for the new indexes (HOMA-beta and Matsuda) have been included in detail in Table 1 and discussed in the text together with the rest of the biochemical variables and the HOMA-IR index. 

Comment 3: Fig. 2B: This figure is still not plausible (to me). Already at baseline, patients in the FMT group (=receivers) seemed to share some gut microbiota with donors (which is plausible; orange bar), but you also claim, that they had other microbiota which were only present in donors (black bar), even though they did not yet receive FMT. So, if the black bar is not about shared species, how can receivers contain species, that are specific to donors and could not have yet reached the receivers? At baseline, there can only be orange and blue bars.

Answer 3: I am grateful for your understanding. We have now taken your position into account and amended the figure accordingly. Please refer to the new figure 2b in the text.

Final comments

We would like to take this opportunity to thank the editorial team once again for allowing us to revise and improve our manuscript. 

We trust that our revised manuscript meets the standards required for publication in the journal. Nevertheless, we are open to further revisions in accordance with any additional suggestions or concerns raised by the editor or reviewers.

Sincerely,
The Authors